# Mitochondrial Dysfunctions: A Red Thread across Neurodegenerative Diseases

**DOI:** 10.3390/ijms21103719

**Published:** 2020-05-25

**Authors:** Serena Stanga, Anna Caretto, Marina Boido, Alessandro Vercelli

**Affiliations:** 1Department of Neuroscience Rita Levi Montalcini, University of Turin, 10126 Turin, Italy; anna.caretto@unito.it (A.C.); marina.boido@unito.it (M.B.); alessandro.vercelli@unito.it (A.V.); 2Neuroscience Institute Cavalieri Ottolenghi, University of Turin, 10043 Orbassano (TO), Italy; 3National Institute of Neuroscience (INN), 10125 Turin, Italy

**Keywords:** mitochondria, cellular homeostasis, neurodegenerative diseases, motor neuron diseases, neurodegeneration, oxidative stress, mitochondria biogenesis and dynamics, mitochondria targeting drugs

## Abstract

Mitochondria play a central role in a plethora of processes related to the maintenance of cellular homeostasis and genomic integrity. They contribute to preserving the optimal functioning of cells and protecting them from potential DNA damage which could result in mutations and disease. However, perturbations of the system due to senescence or environmental factors induce alterations of the physiological balance and lead to the impairment of mitochondrial functions. After the description of the crucial roles of mitochondria for cell survival and activity, the core of this review focuses on the “mitochondrial switch” which occurs at the onset of neuronal degeneration. We dissect the pathways related to mitochondrial dysfunctions which are shared among the most frequent or disabling neurodegenerative diseases such as Alzheimer’s, Parkinson’s, and Huntington’s, Amyotrophic Lateral Sclerosis, and Spinal Muscular Atrophy. Can mitochondrial dysfunctions (affecting their morphology and activities) represent the early event eliciting the shift towards pathological neurobiological processes? Can mitochondria represent a common target against neurodegeneration? We also review here the drugs that target mitochondria in neurodegenerative diseases.

## 1. Introduction: The Peculiarity of Mitochondria

Classically, mitochondria were considered the “powerhouse” of the cell, the energetic core generating ATP for cell activities [1]. Intensive research on their morphology and functions showed that mitochondria play several different roles. 

Interestingly, mitochondria possess their own genome, the mitochondrial DNA (mtDNA), packed in nucleoids in the mitochondrial matrix in close association with the mitochondrial inner membrane [2]. Mutations at the mtDNA level are causative for many human diseases which are generally defined as mitochondrial disorders [3]. Moreover, mitochondria are vital and dynamic organelles able to modify their shape and size in order to respond to the cellular needs and, therefore, to maintain cellular homeostasis [4]. Mitochondria are the site of the oxidation of metabolites, for example through Krebs’ cycle, and of the β-oxidation of fatty acids [5]. Mitochondria are also the main generators of the reactive oxygen species (ROS) [6] and they participate to cell proliferation by maintaining a proper redox state and by recycling oxidized electron carriers. Importantly, mitochondria buffer calcium ions regulate, in turn, calcium homeostasis [7,8]. This is a crucial mitochondrial function impacting many cellular pathways such as the neurotransmitters’ release from neurons and glial cells [9].

In order to maintain the homeostasis of the cell, mitochondria play a crucial role in the choice of cell fate since they are also able to control cellular programmed cell death [10]. Indeed, mitochondria can induce apoptosis via caspase-dependent or independent mechanisms, the first by the activation of pro-apoptotic members of the B-cell lymphoma 2 (Bcl-2) family, the latter by the release of toxic mitochondrial proteins occurring after mitochondrial loss of function [11].

Mitochondria are extremely sensitive to every subtle change perturbing the homeostasis of the cell and, in turn, can modify their shape and number. Indeed, the processes of fusion and fission are fundamental, respectively, to repair a damaged mitochondrion or to increment their number, as in case of an increased demand of energy or to facilitate their removal when damaged in order to protect cellular integrity [12].

Additionally, the number and the size of cristae, which are dynamic bioenergetic compartments of the inner mitochondrial membrane where the respiratory chain occurs, adapt to the needs. Their plasticity guarantees a constant turnover to assure the balance between regeneration, biogenesis, and elimination of damaged mitochondria [13]. On the other hand, a regressive event occurring to mitochondria consists in “mitophagy”, i.e., the process of mitochondria degradation by autophagy [14], aiming to maintain the correct turnover of mitochondria [15]. 

Finally, mitochondria are also able to protect cell integrity by preventing the damage induced by viral infection [16]; indeed, they can also stimulate the innate immune response against these insults [17]. 

## 2. The “Mitochondrial Switch” and Its Impact on Neurodegeneration

Mitochondria are extremely sensitive to the insults that occur and accumulate in the cell, directly impacting on their function, consequently promoting disease development and progression. Indeed, besides human hereditary diseases caused by mutations of the mtDNA or in nuclear genes responsible for mitochondrial deregulation [18], mitochondrial dysfunctions are among the first events which occur in a vast number of pathological conditions ranking from diabetes [19], inflammatory diseases (such as multiple sclerosis [20]), to cancer [21] and neurodegenerative diseases [22]. We review here the “mitochondrial switch” which occurs during neurodegeneration. Neurodegenerative diseases consist of a group of heterogeneous disorders, but, nevertheless, they are all characterized by the progressive loss of specific neuronal populations and circuits in the central nervous system (CNS) triggered by mitochondria dysfunctions [23,24]. Another feature of the majority of neurodegenerative diseases is the progressive and highly disabling motor decline. Indeed, besides neuronal cells, also muscle cells are particularly enriched in mitochondria, since they need a high level of energy to function, and are heavily impacted by mitochondrial dysfunctions. There is growing evidence in the field exploring mitochondrial defects occurring also in the peripheral nervous system (PNS) and peripheral cells. Due to the low potential of neural regeneration, mitochondrial damage results in detrimental effects for neuron survival [25], while in peripheral cells the regeneration capacity is higher and clinical symptoms are evident only later in the disease. 

Globally, in neurodegenerative diseases there is a “switch” in mitochondrial function which contributes importantly to the transition from a normal physiological to a degenerative condition. The accumulation of different stresses and the parallel impairment of a number of cell protective processes elicits neurodegeneration (Figure 1). 

During neurodegeneration oxidative stress increases and intracellular ROS are formed, inducing to mtDNA mutations [1] and also to the disruption of mitochondrial membranes and cristae with a decrease in ATP production. Among the mechanisms eliciting oxidative stress, it is worth mentioning the process of deuteronation. Deuterium is a hydrogen isotope, present in nature in a well-defined ratio with hydrogen (1:6600) [26]: in body fluids, the probability of ATP synthase deuteronation process is 1/15,000 and the increase of this ratio can negatively interfere with ATP synthesis. Moreover, it has also been demonstrated that deuteronation can enhance formation of free radicals, slowing the electron trafficking in the mitochondrial electron transport chain (ETC) [27]. This mechanism has been evaluated in several conditions or pathologies (i.e., cancer, diabetes, aging, depression) and it can be reverted replacing heavy water with deuterium depleted water (DDW). Moreover, reduction of deuterium content by DDW ingestion can rescue long-term memory in Wistar rats restoring altered deuterium/hydrogen ratio and enhancing synaptic activity [28]. Indeed, DDW is able to promote antioxidant enzymes’ expression (SOD) and restore glutathione (GSH) levels mitigating, in turn, ROS formation and preventing fatty acid oxidation [29].

Moreover, the accumulation of altered proteins occurring in many neurodegenerative diseases can determine a decline in mitochondrial biogenesis, impacting on mitochondrial turnover and inducing structural and functional changes in cells. Different studies showed that various proteins involved in the above-mentioned processes and in mitophagy are affected in neurodegenerative diseases determining a dysregulation of these processes [30]. Indeed, mitophagy is required to remove damaged mitochondria and this process is finely tuned to maintain an equilibrate turnover of mitochondria and, from a bigger picture, proper cellular development and differentiation. Described for the first time in 1998 [31], mitophagy has gained considerable interest in the field of neurodegeneration in the last years; however, because of its fine tuning, it is a difficult target for designing new therapies against neurodegeneration. Some authors hypothesized a relation between mitophagy dysregulation and neurodegenerative diseases, based on the impairment in autophagosome–lysosome fusion and defects in lysosomal acidification, often observed in neurodegeneration [32].

Altogether, these events can trigger the process of mitochondrial membrane permeabilization and, when severely injured mitochondria are not appropriately removed, they release their contents into the cytosol and the extracellular environment: this represents a starting point of both apoptosis and necrosis. This process is called generation of damage-associated molecular patterns (DAMPs) and mitochondria represent the major site of DAMP formation. By triggering cell death, mitochondria-derived ROS, calcium, ATP, and the other signaling molecules also lead to neuroinflammation. The process of neuroinflammation refers to the proliferation and activation of microglia and/or astrocytes (microgliosis, astrogliosis) and, to note, neurodegenerative diseases all show evidence of neuroinflammation; the specific disease-related peculiarities are described later on in the sections dedicated to each pathology. Interestingly, neuroinflammation is closely related to mitochondria [33]: indeed, mitochondria can regulate inflammatory signaling and can modulate the innate immunity by generating ROS and, as a consequence, triggering neurodegeneration [34]. Closely related to ROS and DAMP formation, another largely studied mechanism by which mitochondria drive neuroinflammation is their ability to activate the NOD-, LRR-, and pyrin domain-containing protein 3 (NLRP3) inflammasome [35], in turn leading to the release of proinflammatory cytokines [36]. Interestingly, NLRP3 localizes in the cytosol and once activated it translocates into mitochondria and mitochondria-associated membranes (MAMs) [37]. Another mechanism by which mitochondria trigger and sustain neuroinflammation is closely related to their dynamism: indeed, the inhibition/stimulation of fission induces/reduces NLRP3 inflammasome assembly and activation, respectively [38].

In particular, mitochondrial dysfunctions (both affecting neuronal survival and triggering neuroinflammation) have been reported in Alzheimer’s (AD), Parkinson’s (PD), and Huntington’s (HD) diseases, Amyotrophic Lateral Sclerosis (ALS), and Spinal Muscular Atrophy (SMA), thus representing a potential biomarker of the diseases, as one major unifying basic mechanism involved in aging and neurodegeneration [39,40]. In the subparagraphs below we describe the mitochondrial dysfunctions which are peculiar or shared among the above-mentioned diseases. 

### 2.1. Mitochondrial Dysfunctions in AD 

AD is the most common and devastating neurodegenerative disease expected to affect more than 81 million people worldwide by 2040 [41,42]. The first evidence of pathogenic mechanisms in AD has been the progressive accumulation of beta-amyloid peptide (Aβ) extracellularly in the brain and of neurofibrillary tangles of hyperphosphorylated tau inside neurons [43], determining the progressive loss of cortical and hippocampal neurons and inducing brain atrophy, consequent cognitive and memory loss. More recently, advanced neuroimaging techniques showed evident metabolic alterations in the brain of AD patients at a very early stage of the disease [44]. Indeed, the degenerative process probably starts 20–30 years before the clinical onset and one of the major goals of AD research is the early detection before symptom onset to prevent the disease outbreak. Oxidative stress-induced damage occurs even before amyloid and tau deposition, and mitochondrial dysfunctions are early events which precede neurodegeneration [45]. In AD there is also evidence of neuroinflammation; indeed, microglia and reactive astrocytes are closely associated with amyloid plaques in patients’ brains [46].

Mitochondria appear morphologically and functionally altered, impacting on many processes such as excessive ROS formation, and resulting in the decrease of brain energy because of the reduction of ATP [47], alteration of calcium homeostasis, and of apoptosis induction [48]: indeed, altered levels of apoptotic markers such as Bcl-2, Bax, and caspases have been observed in models of AD [49]. Moreover, alterations in mitochondrial dynamics appear early in AD development, contributing to impair neuronal autophagy because of defective mitophagy [50]. There is a vast literature on the detrimental impact of the accumulation of damage and misfolded proteins inducing dysfunctions at the cellular and mitochondrial level not only in the NS but also in peripheral tissues such as patients fibroblasts [51] which show mitochondrial calcium dysregulation similar to those occurring in the brain [52]. Moreover, AD-related proteins, such as the Amyloid Precursor Protein (APP) and Presenilins (PSs), have been recently described as involved in peripheral processes associated to muscle trophism, neuromuscular junction (NMJ) formation and mitochondrial morphology [53,54,55]. Mitochondrial dysfunction in peripheral cells could represent a new target for early and more accessible diagnosis and, eventually, therapy for AD. 

### 2.2. Mitochondrial Dysfunctions in PD

PD, the second most common neurodegenerative disease in aging, is characterized by the reduction of dopamine (DA) levels in the striatum due to the degeneration of dopaminergic neurons in the SNpc. Patients show a progressive muscle rigidity and tremors due to decrease in dopaminergic modulation on striatal neurons altering motor systems [56,57]. Inclusions of Lewy bodies and presence of immunoreactive α-synuclein and ubiquitin are also hallmarks of the disease [58]. Moreover, neuroinflammation plays a role in PD; reactive microglia in the substantia nigra together with positron emission tomography (PET) imaging studies of PD brains provided early evidence for the role of neuroinflammation in PD [59,60].

The first evidence of mitochondrial dysfunction in PD came in 1983, when Langston described chronic parkinsonism effects in young humans after the use of 1-methyl-4phenyl-1,2,3,6-tetrahydropyridine (MPTP), which is a prodrug to the neurotoxin 1-methyl-4-phenylpyridinium (MPP+); the latter interferes with the activity of the complex I of the electron transport chain [61] whose functioning is fundamental for dopaminergic neuron survival. Decades of research in the field confirmed that mitochondrial dysfunctions are a common feature in PD and are present in animal models, in induced pluripotent stem cell (iPSCs)-derived neurons and in patients’ brains [62,63,64,65]. Mitochondrial dysfunctions are related to both sporadic and familial PD and are associated to disturbances of mitochondrial function, morphology, and dynamics (for a detailed review see Bose and Beal [66]). High levels of ROS, due to the high metabolic demand, determine accumulation of toxic oxidative species and structural alterations of complex I, affecting mitochondria functionality in patient brains and mouse models [67,68]. Moreover, α-synuclein oligomers accumulation induces the permeabilization of the mitochondrial membrane and direct toxicity by increasing ROS production and consequently leading to neuronal death [69]. Disturbances in calcium homeostasis and calcium overload could force the opening of the mitochondrial permeabilization transition pore, resulting in ROS formation, Cyt C release, and apoptosis activation [70]. 

Mitochondrial dysfunctions in PD are also a result of changes in mitochondria biogenesis caused by the dysregulation of transcription factors such as the peroxisome proliferator-activated receptor gamma coactivator 1-alpha (PGC1α), as demonstrated both in mice and humans [71,72]. Furthermore, mitochondrial fragmentation, which occurs rapidly after the loss of membrane potential, has been observed in PD: indeed, many proteins involved in fusion and fission (such as the dynamin-related protein 1 (DRP1)) are altered in a model of familial PD [73]. Notably, genes such as PINK1 and Parkin, which are related to familial-PD forms, are involved in the control of mitochondrial dynamics [74] and several other genetic mutations, including PINK1, Parkin, DJ-1, LRRK2, and α-Syn, have been linked to familial PD and the corresponding gene products are also involved in mitophagy. Additionally, also in PD, mitochondrial dysfunctions are not limited to CNS, but also involve peripheral tissues as it has been described in skeletal muscles and platelets [62,75]. 

### 2.3. Mitochondrial Dysfunctions in HD

HD is an autosomal dominant neurodegenerative disorder, caused by the expansion of CAG repeats in exon 1 of the huntingtin gene (HTT), leading to the specific loss of GABAergic striatal medium spiny neurons. Patients show impairments in behavior, muscle coordination, and a progressive mental decline leading to death [76]. The huntingtin protein (HTT) is involved in many processes such as axonal transport, signal transduction, and autophagy; its misfolding due to HTT mutation determines the disruption of the above-mentioned biological processes and of many others [77]. Mitochondrial dysfunctions play a central role in HD progression. Indeed, abnormal oxidative stress and ROS production, calcium imbalance, and decreased enzymatic activity of the respiratory chain complexes resulting in the alteration of lactate production have been observed both in mouse models of HD and in patients’ brains [78,79,80]. Several studies have revealed reduced mitophagy in the brain of HD patients [81]. Moreover, accumulation of mtDNA defects occurs early in HD to the point of being suggested as a potential biomarker of the disease [82,83]. Altogether, these mitochondria-derived defects recall pro-inflammatory activated innate immune cells; indeed, in HD, a remarkable neuroinflammation has been observed in brains from HD patients and PET imaging showed that microglia activation correlates with the pathology progression (for an extensive review on the subject see [84]).

As for the majority of neurodegenerative disorders, the approach in HD research field has always been to focus on the neurological symptoms and phenotypes. However, the HTT gene is ubiquitously expressed [85] and in HD there are important dysfunctions occurring outside the brain. In this regard, impaired uptake and levels of glucose and cardiac dysfunctions have been shown in HD mouse models [86,87]. Moreover, HD patients show a high predisposition to diabetes and muscle wasting, actually before neurodegeneration [88,89]. Interestingly, these dysfunctions are closely connected to mitochondrial activity and to tissues with high mitochondrial density. Moreover, mitochondrial alterations are accompanied by oxidative stress in skin fibroblasts of patients [90]; additionally, the enzymatic activity of the Aco2, which catalyzes the conversion of citrate to isocitrate in the TCA cycle, is impaired in PBMCs of HD patients and PreHD carriers [91]. Authors suggest it as a potential biomarker to assess the disease status of both patients and carriers, being an easy and affordable non-invasive blood test in the clinical routine.

### 2.4. Mitochondrial Dysfunctions in ALS 

ALS is a neurodegenerative disorder due to the loss of upper and lower motor neurons (MNs) which entails muscle atrophy, progressive weakness, and respiratory failure leading to death [92]. Most cases are sporadic, whereas 10% are familial, with mutations mainly occurring on superoxide dismutase 1 (SOD1), FUS, TDP43 genes causing the accumulation of the corresponding mutant proteins [93,94,95]. However, the etiology of the disease is not yet understood: it includes excitotoxicity, protein misfolding and aggregation, dysregulation of RNA metabolism, and neuroinflammation [96]. Inflammatory changes have been largely described in ALS patients: astrogliosis is evident within the spinal cord (both ventral and dorsal horns) and brain (cortical gray matter and subcortical white matter), whereas microgliosis in the spinal cord ventral horns, corticospinal tract, and motor cortex [97,98].

For sure, mitochondrial-dependent dysfunctions such as ATP deprivation, oxidative stress, impaired cell signaling together with altered mitochondrial morphology/dynamics are also correlated to the pathogenesis of ALS and represent very early phenomena [99]. More in details, (i) a high amount of ROS, free radicals, and other toxic species such as peroxynitrite (ONOO^−^) [100] and (ii) an alteration of the levels of expression of proteins involved in mitochondrial fusion and fission (i.e., OPA1, Mitofusin for fusion, and Fis1 and DRP1 for fission) [100], were observed and (iii) apoptotic signaling is impacted in different ALS models. The latter triggers cell death, pivotal to keep the organism healthy by eliminating cells which cannot be rescued from damage, and mitochondria are fundamental for this. Indeed, as previously introduced, mitochondria induce apoptosis either via caspase-dependent or independent mechanisms. In ALS, the apoptotic cascade is activated [101] and the mutant SOD1 is playing a major role in this cascade by interacting with Bcl-2 [102]. Indeed, therapeutic strategies targeting mitochondrial dysfunction to prevent ALS progression represent a possible treatment option [103]. 

As for the other neurodegenerative diseases, also in ALS the PNS is dramatically affected and mitochondrial dysfunctions occur in peripheral tissues as well. MN degeneration is preceded by NMJ denervation, determining a dying-back retrograde neuropathy. Such defects may cause MN death by predisposing them to calcium-mediated excitotoxicity, increasing ROS generation and initiating the apoptotic pathway [104]. Moreover, features of mitophagy in the presynaptic terminals of ALS’ NMJs have been observed [105], suggesting impaired mitophagy as one of the mechanisms causing degeneration (for a review see Elfawy and Das [106]).

### 2.5. Mitochondrial Dysfunctions in SMA

SMA affects MNs in children and young adults with a mutation/deletion of the Survival Motor Neuron 1 (SMN1) gene [107]. SMN has a role in the assembly of small ribonucleoprotein particles, that function in pre-mRNA splicing and gene transcription and in mRNA transport in MN axons [108]. The outcome of its genetic alteration is a decrease in the levels of the functional protein which results in motor impairment, muscle atrophy, and premature death [109]. The disease severity in humans depends on SMN2 which is a highly homologous gene of SMN1. However, SMN2 produces only 10% of fully functional SMN; therefore, the degree of compensation relies on patient’s SMN2 copy numbers. Although the genetic cause of SMA has been identified, many aspects of its pathogenesis remain unclear. Interestingly, SMN deficiency has been associated to oxidative stress, mitochondrial dysfunction, and impairment of bioenergetic pathways in different models. SMN silencing in NSC-34 cell lines induces an increase in cytochrome c oxidase activity and mitochondrial membrane potential, generating free radicals [110]; in human SMA iPSCs, axonal mitochondrial transport and mitochondrial number and area are altered [111]. Studies in SMA murine models also show alterations in mitochondrial respiration, mitochondrial membrane potential and mobility, and confirm an increased oxidative stress level and fragmentation [112]. Additionally, proteomic studies show that bioenergetic pathways associated to glycolytic enzymes, such as GAPDH and PGK1, are altered in SMA models [113,114]. Moreover, besides cell-autonomous motoneuronal toxicity, also glial-mediated inflammation seems to negatively impact on neuronal survival and to promote progression and propagation of the degenerative process by the activation of apoptotic cascades [115].

Even if SMA has been classically categorized as a MN disease, SMA pathogenesis is more complex and many mechanisms and districts are involved. Indeed, since SMN protein is ubiquitously expressed, its lack affects not only MNs, but also NMJs, which show dramatic alterations, including immaturity, denervation, neurofilament accumulation, and impaired synaptic functions [116]. Additionally, skeletal [117] and heart muscle cells [118] are also dramatically affected: they require high levels of energy and are, therefore, particularly enriched in mitochondria. Alterations in the process of mitophagy have been described in the muscles of SMA patients [119]. Some researchers suggest that these peripheral abnormalities may be the origin of the detrimental effects observed on MN survival through retrograde signals coming from the muscles/NMJs [120,121]. On the contrary, others support the idea that peripheral dysfunctions derive from MN degeneration which, in turn, results in impairment of the nerve-muscle interplay [122]. This ‘chicken or the egg’ question about where the degeneration starts could help with understanding the important mechanisms of the disease and the development of disease-modifying interventions which could have greater therapeutic impact by preferentially targeting the CNS, the PNS, or peripheral tissues. Above all, one possibility is that mitochondrial impairment, which is an early event in SMA, occurs both at central and peripheral level, probably representing a relevant target independently from the tissue. 

## 3. Are Mitochondria the Red Thread in Neurodegenerative Diseases? 

Considering the enormous economic and social impacts, finding a cure for neurodegenerative disorders remains a priority in science. Researchers are focused on identifying the common pathogenic processes shared among these diseases, in order to design new treatments and/or drug combinations and repurposing. Indeed, scientists and pharmaceutical companies aim at the development of common therapies for multiple neurodegenerative diseases. To this aim, understanding the co-occurrence and overall interactions among these diseases is the first step for drug development. 

The majority of neurodegenerative disorders have significant genetic components, with genetic heritability such as for AD, PD, HD, ALS, and SMA. To date, the most common and methodologically defined approaches used are based on microarrays, next-generation RNA sequencing, statistical and bioinformatics methods to detect shared genes, single nucleotide polymorphisms (SNPs), and profile networks of pathways underlying neurodegenerative mechanisms [123,124,125,126]. Genome wide association studies, which represent the largest publicly available resource in the genomic domain, reveal the similarities among diseases and can help in the process of drug screening and discovery (for a comprehensive review see Arneson et al. [127]). In addition to genetics, considering that the majority of the forms of neurodegeneration are sporadic, factors such as the ageing of the global population, the lifestyle, and the exposure to chemicals are taken into account too and are nowadays known risk factors for neurodegeneration [128,129]. 

Neurodegeneration can be defined as a process characterized by the loss of neuronal populations, progressive cognitive decline, and/or motor symptoms. As largely described above, there is overwhelming evidence of impaired mitochondrial functions as a causative factor driving the development of neurodegenerative diseases together with the unavoidable elevation of oxidative stress, the increase of free radicals in the brain, impaired DNA repair capability, and decreased tissue regeneration. Indeed, besides the intrinsic differences among diseases, mitochondria are a shared key crossing point in the biological processes driving neurodegeneration and determining the clinical outcomes. Mitochondrial dysfunctions could be considered a possible red thread in neurodegenerative diseases. This is the consequence of the fact that mitochondria, besides being the powerhouse of eukaryotic cells, are fundamental organelles involved in critical pathways connected to cell growth and differentiation, cellular signaling, apoptosis, and cell cycle control. Together with the fact that mitochondrial dysfunctions appear at the early disease onset and contribute to disease progression, they are a hallmark of neurodegeneration. 

Interestingly, the mitochondrial switch driving neurodegeneration occurs indistinctly in every tissue, even in the periphery although neurodegeneration has always been approached and studied focusing on the CNS. To some extent, one could investigate neurodegenerative disorders as affecting mitochondria primarily. In Figure 2 the mitochondrial dysfunctions and effects on the CNS, PNS, and periphery are represented for AD, PD, HD, ALS, and SMA. Therefore, targeting mitochondria can represent new therapeutic approaches for different neurodegenerative diseases. Below, molecules tested for neurodegeneration targeting the main pathways related to mitochondrial-dependent dysfunctions, such as oxidative stress, mitochondrial biogenesis, mitochondrial membrane permeability and dynamics, are listed.

## 4. Therapies Targeting Mitochondria 

Despite different time of onset, symptoms, and etiology, the above-mentioned neurodegenerative pathologies share the progressive neuronal degeneration [130,131], resulting in the loss of neuronal populations and impairment of neurotransmission (a highly energy-demanding process) [132]. However, even if these neurodegenerative diseases have been studied for decades, up to now, most of them lack of disease-modifying therapies: the available treatments (summarized together with their main effects and limitations in Table 1) are generally symptomatic and they are unable to fully prevent the disease progression [133,134,135,136,137,138]. 

Nevertheless, in the last years the landscape of the potential drugs has expanded, in the attempt not only to stop the disease progression, but also to prevent the onset of symptoms [139]. To this aim, the identification of an early and possibly common target could represent a turning point in the treatment of these pathologies: in this scenario, mitochondria could be considered promising targets, because (i) their impairment can be detected since the earliest stages, influencing the onset and progression of the diseases, (ii) their dysfunctions are common to all the pathologies described here [140,141]. In the next paragraphs we will focus on the most common therapies that target these organelles in neurodegenerative diseases and that have been object of preclinical studies and/or whose role has provided promising outcomes in clinical trials, classifying them according to the mechanisms of action.

### 4.1. Antioxidant 

As previously mentioned, ROS are produced by mitochondria both in physiological and pathological conditions [149] and can be responsible for onset and progression of many neurodegenerative diseases [150,151]. Furthermore, as discussed above, one of the major events related to neurodegenerative diseases, driven by the redox status, is microglial activation-derived neuroinflammation [152]: the induction of the expression of proinflammatory genes leading to the release of cytokines and chemokines can represent a consequence of the uncontrolled ROS production by mitochondria. This chronic inflammatory state is characteristic of many neurodegenerative diseases [153].

Many synthetic or natural molecules (summarized in Table 2) have been investigated, since they can reduce the ROS-induced effects, among which neuroinflammation, in a similar way due to their molecular structure, through direct or indirect mechanisms. 

#### 4.1.1. Synthetic Antioxidants

As reviewed by Cenini and Voos [139], some compounds such as MitoQ play a key role in the ETC preventing ROS formation due to the electron leakage to oxygen. It has a lipophilic tail that lets it cross the inner mitochondrial membrane (IMM) and a benzoquinone ring that can be reduced to ubiquinol and oxidized back to ubiquinone, performing the antioxidant activity [154,155]. It induced neuroprotective effects in preclinical studies in vitro on AD cortical neurons and determined an increased lifespan and a milder cognitive decline, due to cholinergic neurons protection and synaptic preservation in AD mice [139,155]; furthermore, MitoQ prevented PD mice dopaminergic cells loss [156,157], increased lifespan and hindlimb strength of SOD1^G93A^ mice [158], and finally mitigated muscle wasting in R6/2 HD mice [159]. 

As reviewed by Weissig [160], SkQ1 belongs to the quinone derivative group and it is still analyzed only in preclinical studies in OXYS rats providing learning and memory enhancing and behavioral improvement [139,161]. SkQ1 and MitoQ share the TPP^+^, a lipophilic cation that facilitates molecules targeting to mitochondria; the same residue is also present in Mito-Apo, another compound that displayed its neuroprotective effects in AD and PD both in vitro and in vivo providing, respectively, a reduction of neuronal degeneration [162] and an attenuation of motor deficit in PD mouse models when bound to apocynin, a NADPH oxidase inhibitor that counteracts ROS formation [158,163,164]. 

Beyond quinone structures, there are many other aromatic molecules that can provide themselves not only redox activity but also lipophilic properties to cross the membrane. For instance, the phenothiazine ring allows Methylene Blue both to reach the mitochondrial matrix and to perform redox reactions reducing ROS formation [165,166]; indeed, this activity is ensured in vivo, increasing attentional functions and preserving dopaminergic neurons in a PD rat model [157,167]. In Szeto-Schiller tetrapeptides the tyrosine or dimethyl tyrosine residues can scavenge ROS but, differently from the previous drugs, they act in the IMM without joining the mitochondrial matrix, as described by Rocha et al. [168]; both in in vitro and in vivo studies they have shown multiple functions deriving from the antioxidant activity, making it a promising compound for AD therapy [139], but they have also provided interesting results in further studies on ALS cell models and in PD and ALS mice, ensuring increased survival and motor performance enhancement due to neuroprotective effects [169].

Not only an aromatic structure can perform antioxidant activity scavenging ROS: thanks to its dithiolane ring, α-Lipoic acid [170] can perform redox reactions promoting a reduction of cytotoxic events in vitro and an amelioration of cognitive functions in AD animal models; the first promising results on AD patients have been already obtained in association with other antioxidant compounds but, unfortunately, there are no clinical trials to evaluate efficacy of isolated α-Lipoic acid [171].

Unlike the aforementioned compounds, inosine plays an indirect antioxidant activity: its derivative, urate, exerts neuroprotective effects mediated by the nuclear factor erythroid 2-related factor 2 (Nrf2), one of the main controllers of the response to oxidative stress; moreover, it seems to be able to increase GSH levels and its release. This was highlighted in different experimental studies in vitro but positive outcomes have also emerged in PD mouse models, encouraging scientists to consider preclinical studies also on ALS models [172,173]. Nowadays, there are some clinical trials that have confirmed safety and efficacy of inosine in increasing urate levels in both pathologies (NCT00833690; [174]; NCT02288091; [175]).

Finally, as reviewed by Reiter et al. [176], melatonin is able to counteract oxidative stress not only directly interacting with free radicals [177], but also by stimulating antioxidant enzymes and upregulating GSH synthesis, through an indirect pathway. For these reasons, its use in the treatment of neurodegenerative diseases has been proposed: in vitro it exerted an antiapoptotic activity and in vivo it improved behavioral and cognitive functions and extended the lifespan of AD and ALS mice and rats [151,178,179,180,181]; furthermore, it enhanced locomotor performances in PD mouse models [182]. Otherwise, meta-analysis and clinical trials on AD and PD patients revealed only an improvement in sleep quality without any ameliorations in terms of cognitive or motor functions and clinical trials testing melatonin in a specific range dose to establish its neuroprotective activity are needed [181,182,183]. Similarly, N-Acetylcysteine (NAC) can exert both a direct and an indirect antioxidant effect, reacting with ROS or restoring GSH levels [184,185,186]. Its efficacy has been already evaluated in vitro and in vivo, where it, respectively, downregulated apoptotic markers in AD and PD cells and enhanced mitochondrial activity (for example increasing brain connections or reducing lipid peroxidation [186]) and preventing the degeneration of MNs derived from SMA iPSCs [118]; furthermore, a mitigation of cognitive and motor impairment has been observed on HD mice [187]. Finally, different clinical trials have examined its efficacy on AD patients showing an amelioration in cognitive and behavioral functions (NCT01320527; [188]) and an increase in GSH brain levels in PD patients, although this outcome has not been obtained in all clinical trials, maybe because of NAC low oral bioavailability [186]. 

#### 4.1.2. Natural Antioxidants

Beside the synthetic antioxidants, several natural compounds (taken as dietary supplements) can play a direct antioxidant role. Among them, Vitamin C can scavenge free radicals when oxidized to dehydroascorbate and recycled back to ascorbic acid through a radical mechanism [189]; its effectiveness in reducing apoptotic process and in regulating ROS balance and oxygen consumption has been already proven, respectively, on Aβ1-42 peptide-treated human cortical neurons and in APP/PSEN1 and 5XFAD Tg mice [190,191,192]. Even Vitamin E, that includes its tocopherol and tocotrienol derivatives, can regenerate tocopherol structure after lipid peroxidation inhibition, in a mechanism that can involve Vitamin C or GSH, as described by Stahl and Sies [189]. Preclinical studies on aged mice and on APP/PS1 mice showed promising results; indeed, ROS scavenging can recover mitochondrial damage, but also enhance mitochondrial biogenesis and bioenergetics: this in turn has effects on cognitive and behavioral impairment improvement [193,194,195]. Despite these encouraging outcomes in preclinical stages, further clinical trials are needed to investigate the real effects of Vitamin E supplementation in AD in order to consider all the factors that can influence patients’ responsiveness [196]. Even in ALS, Vitamin E intake is debated because the positive outcomes obtained in preclinical trials on ALS onset and progression cannot always be translated to patients [151,197].

Carotenoids (i.e., Astaxanthin) can exert a double antioxidant activity, since they can both quench ROS through a radical mechanism (for an extended review see Fiedor and Burda [198]), or improve the activity of several antioxidant enzymes like SOD, CAT, and GPX [199]; for instance, their efficacy has been already proven on an AD cell model showing a prevention of synaptotoxic events due to ROS production [200]. 

### 4.2. Mitochondrial Biogenesis and Permeability

Mitochondrial biogenesis is an important process, since increasing mitochondria number can partially rescue, at least from a quantitative point of view, the low efficiency of the impaired ones, through a compensatory mechanism [201]. In this scenario, PGC-1α represents the main regulator in mitochondrial biogenesis (Table 3). Its reduction has been related to PD onset, because it could indirectly induce alpha-synuclein oligomerization: in fact, it has been proven that PGC-1α restoration can reduce alpha-synuclein toxicity both in cell culture and in transgenic animals [202]. To further maintain mitochondrial functionality, the prevention of the formation of the mitochondrial permeability transition pore (mPTP) could be a good strategy: it can avoid mitochondria swelling, outer membrane disruption, and releasing of apoptotic factors [203], finally resulting in neuronal death. According to this, Olesoxime is a molecule that can bind two proteins of the outer mitochondrial membrane (TSPO and VDAC), in this way preventing the mPTP opening [204]. Following different preclinical investigations in which it provided lifespan prolongation in transgenic mouse models [204], this drug has been considered a promising compound for ALS and SMA patients but, even if clinical trials on ALS have failed [205], a Phase 2 trial on SMA patients has shown an amelioration in motor functions (NCT01302600) that encouraged further investigations [206]. Moreover, some studies on PD and HD models have confirmed both in vitro and in vivo its key role in mitochondria stability resulting in a mitigation of cognitive and behavioral impairment [206]. 

### 4.3. Mitochondrial Bioenergetics 

Neurotransmission and metabolism require high energy levels, and mitochondria are the key organelles for ATP production; therefore, acting on impaired mitochondria with compounds that may support neurons in carrying on their activity could be a good strategy in neurodegenerative diseases treatment (Table 4). 

As exhaustively explained by Carrera-Juliá et al. [151], NAD+ is a compound involved in many metabolic pathways that are related to neuroprotection and, in turn, its intake could be considered a promising approach to treatment of neurodegenerative diseases. Indeed, acting in glycolysis, Krebs cycle, and oxidative phosphorylation, it can contribute to preserve cognitive functions, representing a promising compound for AD treatment [139,207]. Furthermore, the role of Nicotinamide Riboside, a NAD+ precursor, has been investigated in vivo in AD models and in vitro in PD and ALS mutated cells, providing a wide range of positive outcomes, from mitochondrial biogenesis enhancement to ROS production inhibition, reflected in cognitive improvement due to an increase of synaptic plasticity for what concerns AD mice [64,151,208].

Similarly, triheptanoin can provide several metabolites that powers Krebs cycle, both in CNS and in peripheral tissues [209]. Therefore, its anaplerotic role has been investigated at early stages of HD: however, until now, only one clinical study has confirmed such efficacy, showing an improvement in patients’ brain metabolic profiles (NCT01882062; [210]). Furthermore, thanks to its function, triheptanoin significantly delayed motor neuron loss and motor symptom onset in SOD1^G93A^ mice [211]: for this reason, it has been also considered as a promising treatment to slow ALS progression.

### 4.4. Compounds Targeting Multiple Mitochondrial Dysfunctions

Besides all the compounds listed until now, there are several other molecules which may act at the same time on many pathways linked to mitochondrial dysfunctions (for a general overview see Table 5). For example, phenylpropanoids are natural molecules derived from the amino acid l-phenylalanine; among them, resveratrol, curcumin, and flavonoids (including epigallocatechin-gallate, quercetin and wogonin) can (i) regulate mitochondrial biogenesis, enhancing the expression of several activators (i.e., PGC-1α, SIRT1, Nrf1, Nrf2, and TFAM), (ii) improve mitochondrial bioenergetics, (iii) enhance ATP production, and (iv) counteract apoptotic pathways (as described in Kolaj et al. [212]). Furthermore, they are able to prevent oxidative stress (i) quenching free radicals through a direct mechanism [213], (ii) upregulating the expression of antioxidative enzymes via the activation of different signaling pathways (i.e., Nrf2), even increasing GSH production [214,215], and (iii) inhibiting ROS-forming enzymes like XO and NOX [216]. Therefore, all these compounds have been investigated as potential therapeutics to treat mitochondrial dysfunctions characterizing many neurodegenerative diseases. When administered in vitro in AD and ALS cell lines, these compounds can significantly increase cell survival, reducing oxidative stress and restoring MMP [139,151,217,218,219]; likewise, preclinical studies in vivo on AD and ALS models confirmed their neuroprotective effects, improving cognitive functions and delaying motor symptom onset and disease progression, respectively [73,139,151,220]. Furthermore, the first promising results on human clinical trials have appeared: in fact, curcumin was effective in lifespan elongation and in decreasing disease progression in ALS patients, although the low oral bioavailability makes new delivery methods necessary [151]; and resveratrol has shown to be safe, tolerable, and effective in mitigating cognitive decline in clinical trials on AD patients (NCT01504854 [221]; NCT00678431 [222]).

Other compounds, derived from drug repurposing studies, are extremely effective in mitigating mitochondrial alterations, although initially employed for different therapeutic purposes and acting on different pathways. Among them, pramipexole, a dopamine receptor agonist approved for PD therapy, seems to have an antioxidant role independent from dopaminergic receptors activation [223]: indeed, it plays a key role in neuroprotection, preventing ROS formation and consequently reducing cell death, as proven on SH-SY5Y cells exposed to MPP+ [157,224]. Moreover, as demonstrated through a patch-clamp experiment on rat liver and brain mitochondria, pramipexole may also inhibit the mPTP opening, preventing mitochondria Ca^2+^-dependent swelling [225]. 

Another compound investigated for PD treatment is the Selegiline-analog N-Methyl,N-propynyl-2-phenylethylamine, a MAO-B inhibitor. Besides ameliorating motor impairments of MPTP-treated mice, it was also able to (i) increase complex I activity and UCP-2 expression (reducing oxidative stress) and (ii) to inhibit p53 mitochondrial translocation (maintaining mitochondrial membrane integrity and counteracting apoptosis) [226].

Other molecules, which are not specific for the treatment of neurodegenerative diseases, can counteract mitochondrial alterations, preventing a wide range of altered mechanisms. For instance, 3-N-butylphthalide (NBP), a compound widely employed in the treatment of ischemic stroke in China thanks to its ability in restoring microcirculation [227], can (i) maintain mitochondria permeability and dynamics (preventing mitochondria swelling), (ii) improve mitochondria bioenergetics (increasing complex IV activity), (iii) prevent cytochrome c release and caspase-dependent apoptosis, and (iv) counteract ROS production. Indeed, most of these effects have been observed on in vitro and in vivo PD models, determining neuroprotection and behavioral improvement [228,229] and a randomized controlled trial has demonstrated NBP efficacy in improving motor symptom and sleep quality of PD patients [230]. Further studies on cellular models of AD have confirmed neuroprotection and this positive outcome has been translated also in vivo determining cognitive enhancement [228]; furthermore, increased lifespan and motor performances amelioration have been observed even in ALS mice thanks to a mitigation of motor neuron loss [228]. 

Finally, several preclinical trials have been performed on ursodeoxycholic acid and tauroursodeoxycholic, two drugs currently approved for the treatment of primary biliary sclerosis [231]. As described by Ackerman and Gerhard [232], although their application in neurodegenerative diseases has to be further investigated, these molecules can (i) exert an antioxidant activity, (ii) stabilize mitochondrial membrane potential, and (iii) downregulate apoptotic markers. Indeed, in vitro they have confirmed their role in apoptosis inhibition and membrane stabilization and in vivo studies have shown behavioral improvement in AD, PD, HD, and ALS models [157,233,234,235]. Nevertheless, at present, positive outcomes in clinical trials have been obtained only in ALS subjects providing a delay in the progression of the pathology (NCT00877604 [236,237]). 

## 5. Conclusions

Besides the recent development of therapies for SMA [238], which still need to be followed for the long-term response, there is no cure for the neurodegenerative diseases that we described in the review. Decades after the discovery of the main pathways and genes involved in the pathophysiological mechanisms of many neurodegenerative processes, researchers and pharmaceutical companies are still investigating a treatment to cure AD, PD, HD, and ALS. We believe that the key for finding new effective targets should be to focus on earlier-phenomena occurring, possibly, before evident degeneration and which could be directed not only to the CNS but also to the PNS and peripheral tissues. Mitochondrial alterations are one of these early and diffuse events. Indeed, mitochondrial dysfunctions appear precociously in neurodegeneration and their accumulation trigger or aggravate the degenerative cascade affecting neuronal tissue, MNs, blood cells, fibroblasts, NMJs, and skeletal muscles. Moreover, mitochondrial dysfunctions represent a common feature in neurodegeneration and should be approached as the crosstalk between AD, PD, HD, ALS, and SMA. 

Recent therapeutic strategies ranging from natural or synthetic compounds to stem cell therapy, are targeting mitochondrial pathways which are dysfunctional in the early phases of neurodegeneration. As we described above, many of them seem promising. 

Looking forward, we expect that some of these targets of mitochondrial-directed therapies could become also biomarkers of neurodegeneration. On the one hand they could be useful to monitor the progression of the disease, on the other hand they could be an effective read-out to test treatment efficacy. The greatest advantage for both the daily practice of clinicians and for the health systems would be the possibility to validate a diagnosis with a non-invasive and cheap method instead of a time, money, and energy consuming long battery of tests. Moreover, the possibility to develop simple and non-invasive tests i.e., from patients’ fluids, would be much more helpful for the amelioration of patients’ motivation and compliance. 

## Figures and Tables

**Figure 1 ijms-21-03719-f001:**
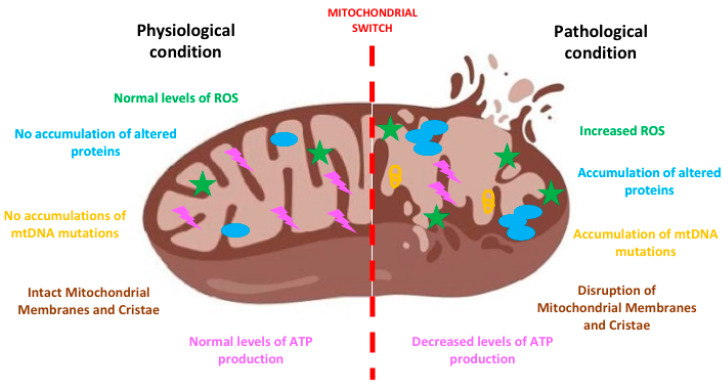
The mitochondrial switch. The sketch shows the appearance and the features of a mitochondrion before and after the accumulation of damage responsible for its impairment. At the onset of the neurodegenerative process the increase of reactive oxygen species (ROS), mitochondrial DNA (mtDNA) mutations and altered proteins determines the swelling of mitochondria and the disruption of their membranes and cristae. Such an altered morphology heavily impacts on function determining, among others, decreases ATP production, increased ROS, leading to neuronal death. Created with BioRender software.

**Figure 2 ijms-21-03719-f002:**
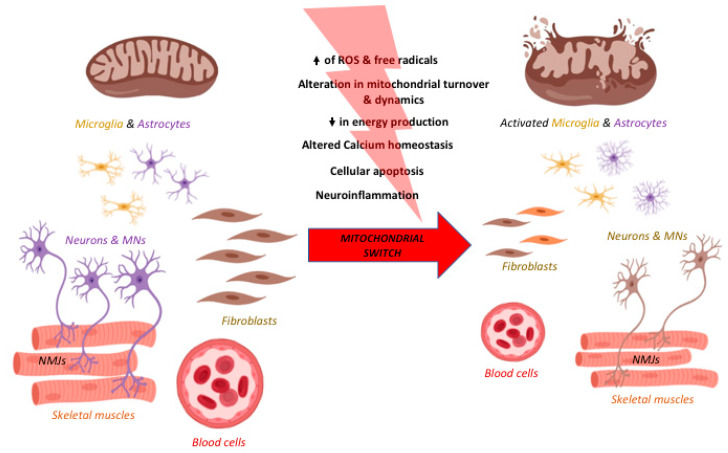
The process of neurodegeneration in the central nervous system (CNS), in the periphery and across diseases. Neurodegeneration is a progressive process taking place not only in the CNS but also in the periphery. After the accumulation of damage at the cellular and, in particular, mitochondrial level, the switch from physiology to pathology is fast and rarely reversible and it is occurring in many cell types. Indeed, it occurs in the CNS but also in the periphery, in particular the neuromuscular junction (NMJ), the skeletal muscles, blood cells, and fibroblasts. Moreover, the major pathways involved are linked to ROS and free radical formation, alterations in ATP formation, calcium homeostasis, apoptosis, mitochondrial turnover and dynamics, and neuroinflammation in terms of microglia and astrocytes activation. These pathways are affected in all the neurodegenerative diseases discussed in this review. Increase of ROS is represented by the black arrow pointing up and reduction in energy production by the black arrow pointing down. Created with BioRender software.

**Table 1 ijms-21-03719-t001:** Available treatments for Alzheimer’s disease (AD), Parkinson’s disease (PD), Huntington’s disease (HD), Amyotrophic Lateral Sclerosis (ALS), and Spinal Muscular Atrophy (SMA). The table summarizes the most common therapies for these neurodegenerative pathologies. Unfortunately, the majority of these treatments are only symptomatic or palliative and cannot stop the disease progression (especially for AD, PD, HD, and ALS) or are less effective in milder and late treated patients (as in the case of SMA).

Pathology	Drug	Mechanism of Action	Main Effects	Main Limitations	References
**AD**	Donepezil, Galantamine, and Rivastigmine	Cholinesterase inhibitors	Acetylcholine increase at synaptic level	Low CNS selectivity; high doses cause gastrointestinal toxicity	[135,142,143,144]
	Memantine	Noncompetitive NMDA antagonist	Reduction of neuronal dysfunctions due to glutamate downregulation	Low beneficial effects in clinical trials (maybe due to late administration)	
**PD**	Levodopa + Carbidopa/Levodopa + Benserazide	DA precursor + DOPA decarboxylase inhibitor	SNC DA level increase	Ineffective in mitigating some motor and non-motor symptoms; dyskinesia	[138,145,146]
	Levodopa + Carbidopa + Entacapone	DA precursor + DOPA decarboxylase inhibitor + COMT inhibitor	SNC DA level increase	Motor fluctuations; dyskinesia	
	Pramipexole and Apomorphine	DA agonists	Activation of DA receptors	Less effective than Levodopa; dyskinesia; expensive	
	Selegiline, Rasagiline, and Safinamide	MAO-B inhibitors	Prevention of DA metabolism	Mild efficacy in monotherapy	
	Amantadine	Antiviral drug	Bradykinesia, tremor and rigidity mitigation, and Levodopa-induced dyskinesia reduction	Several side effects such as hallucination, confusion, blurred vision, and edema	
	Trihexyphenidyl	Anticholinergic	Tremor reduction	Mild effects on motor symptoms	
**HD**	Tetrabenazine (XENAZINE^TM^) and Deutetrabenazine (AUSTEDO^TM^)	Vesicular monoamine transporter (VMAT) type 2 inhibitor	Treatment of pathology- associated chorea (synaptic DA reduction)	Only symptomatic treatment	[134]
**ALS**	Riluzole	Glutamatergic transmission blocker	Antiexcitotoxic effects	Effectiveness limited to the first six months of therapy	[137,147]
	Edaravone (RADICAVA^TM^)	Antioxidant	Free radical scavenger (neuroprotection)	Prescription only for limited cohort of patients	
**SMA**	Nusinersen (SPINRAZA^TM^)	ASO acting on *SMN2* pre-mRNA splicing	Increase of full-length SMN production	Limited efficacy in milder or late-treated patients; expensive; repetitive intrathecal injections	[133,136,148]
	Onasemnogene Abeparvovec (ZOLGENSMA^TM^)	Smn1 delivering by the adeno-associated virus AAV9 (only FDA approved, not yet in therapy)	Increase of full-length SMN production	Limited efficacy in milder or late-treated patients; expensive	

AAV = adeno-associated virus; ASO = antisense oligonucleotide; COMT = catechol-O-methyltransferase; DA = dopamine; FDA = Food and Drug Administration; MAO = monoamine oxidase; NMDA = N-methyl-D-aspartate; VMAT = vesicular monoamine transporter.

**Table 2 ijms-21-03719-t002:** Antioxidant compounds for neurodegenerative disease treatment. The table lists synthetic and natural antioxidant compounds tested for the treatment of neurodegenerative diseases both in preclinical and clinical studies. The most recent and promising studies are collected here. For clinical trial ID we referred to https://clinicaltrials.gov.

Therapeutic Function	Drug / Molecule	Pathology	Preclinical Studies / PMCID	Preclinical Results	Clinical Trials / Trial ID	Clinical Results
**Synthetic antioxidant**	α-Lipoic acid	AD	Preclinical in vitro and in vivo PMC6914903 [171]	In vitro studies: mitigation of cytotoxic effects (reduction of ROS production and lipid peroxidation).	Clinical trials information is collected here PMC6914903 [171]	Safety and neuroprotection are confirmed in combination with other antioxidants conventional treatments but further studies on interactions between them are needed. Isolated α-lipoic acid activity has to be tested
				In vivo studies: memory and learning improvement		
	Inosine	ALS	Under evaluation		Clinical trial Phase 1 Completed, PMC6292193 [175] NCT02288091	Safety, tolerability, and efficacy in increasing urate serum levels
	Inosine/Urate	PD	Preclinical in vitro and in vivo PMC5233635 [173]	In vitro: neuroprotection (Nrf2 transcription and nuclear translocation; GSH increasing)	Clinical trial Phase 2 Completed, PMC3940333 [174] NCT00833690	Safety, tolerability, and effectiveness in increasing urate serum levels
				In vivo: behavioral improvement; reduction of dopaminergic neurons loss		
	Melatonin	AD	Preclinical in vitro and in vivo PMC6826722 [181]	In vitro: protection from apoptosis and neuroinflammation.	Meta-analysis of controlled trials information is collected in the following work PMC6826722 [181]	Improvement in sleep quality but no ameliorations in cognitive functions when melatonin is administered not in combinations with other AD treatments
				In vivo: improvement in cognitive functions and behavioral activities (reduction of neuronal death and beneficial effects on synapses), protection against neuroinflammation		
	Melatonin	ALS	Preclinical in vitro and in vivo PMC7016185 [151]	In vitro: apoptosis inhibition	Clinical safety trial information is collected in the following work PMID: 22739839 [183]	Safety, improvement of sleep quality, and reduction of oxidative stress biomarkers. Further studies to confirm its efficacy alone or combined to other drugs different from Riluzole are needed
				In vivo: survival extension and delay in disease progression (oxidative damage reduction and protection against neuroinflammation)		
	Melatonin	PD	Preclinical in vivo PMC6646522 [182]	Reduction of locomotor deficit (downregulation of lipid peroxidation and dopaminergic cells loss) and of neuroinflammation	Clinical trials information is collected in the following work PMC6646522 [182]	Improvement in sleep quality but no benefits on motor activity
	Methylene Blue	PD	Preclinical in vivo PMID: 30219247 [157]	Attentional functions and motor improvement and neuroprotection		
	Mito-Apo	AD	Preclinical in vitro PMC5392427 [162]	Mito-Apo on dopaminergic neuronal cell line, mouse primary cortical neurons, and a human mesencephalic cell line: reduction of neuronal degeneration and of neuroinflammation		
	Mito-Apo	PD	Preclinical in vitro and in vivo PMC4995106 [163] PMC5651937 [164]	In vitro: neuroprotection against oxidative stress		
				In vivo: motor deficit and neuroinflammation attenuation (neuroprotection)		
	MitoQ	AD	Preclinical in vitro and in vivo PMC6716473 [139]	In vitro: neuroprotection against oxidative stress and neurites outgrowth		
				In vivo: mitigation of cognitive decline and elongation of lifespan		
	MitoQ	ALS	Preclinical in vivo PMID: 24582549 [158]	MitoQ increases hindlimb strength and promotes lifespan elongation of SOD1^G93A^ mice		
	MitoQ	HD	Preclinical in vivo PMC6970224 [159]	MitoQ on R6/2 HD mouse model: reduction of ROS-induced autophagy		
	MitoQ	PD	Preclinical in vivo PMID: 29842922 [156]	MitoQ prevents dopaminergic neurons loss in a 6-OHDA PD mouse model promoting mitochondrial fusion		
	N-Acetylcysteine	AD	Preclinical in vitro and in vivo PMC6320789 [186]	In vitro: apoptosis inhibition and protection against neuroinflammation	Clinical Trial Phase 2 Completed PMID: 25589719 [184] NCT01320527	Cognitive and behavioral improvement
				In vivo: increase of brain connections, GSH levels, TH and Complex 1 activity and protection against neuroinflammation		
	N-Acetylcysteine	HD	Preclinical in vivo PMC3967529 [187]	Cognitive and motor deficits improvement		
	N-Acetylcysteine	PD	Preclinical in vitro and in vivo, PMC6320789 [186]	In vitro: apoptosis inhibition	Clinical trials information is collected in the following work PMC6320789 [186]	Increase of GSH brain levels
				In vivo: increase of GSH levels and reduction of lipid peroxidation		
	N-Acetylcysteine	SMA	Preclinical in vitro PMC4728333 [119]	NAC on iPSCs: mitigation of motor neuron degeneration (increasing in mitochondrial number and axonal transport, reduction of axonal swelling, and apoptosis inhibition)		
	SkQ1	AD	Preclinical in vivo PMC6716473 [139]	Cognitive and behavioral improvement (reduction of ROS formation, improvement of mitochondrial biogenesis and bioenergetics and mitochondrial structure protection)		
	Szeto-Schiller tetrapeptides	AD	Preclinical in vitro and in vivo PMC6716473 [139]	In vitro: mitochondrial biogenesis, bioenergetics and dynamics improvement, and apoptosis inhibition		
				In vivo: anterograde axonal transport and synaptic activity enhancement		
	Szeto-Schiller tetrapeptides	ALS	Preclinical in vitro and in vivo PMC4267688 [169]	In vitro: mutant cells apoptosis inhibition		
				In vivo: increase of survival and behavioral improvement in SOD1^G93A^ mice (neuroprotection)		
	Szeto-Schiller tetrapeptides	PD	Preclinical in vivo PMC4267688 [169]	Lifespan extension and motor performances improvement (neuroprotection)		
**Natural antioxidant**	Carotenoids (Astaxanthin)	AD	Preclinical in vitro PMC4791503 [200]	Astaxanthin on Aβ1-42 oligomers-treated hippocampal neurons: protection against ROS production reducing synaptotoxic events and neuroinflammation		
	Vitamin C	AD	Preclinical in vitro, PMID: 12592670 [190], and in vivo PMC5623070 [192] PMC3944243 [191]	In vitro: inhibition of apoptosis due to mitochondrial membrane depolarization and DNA fragmentation		
				In vivo: Preservation of mitochondrial morphology (attenuation of oxidative stress damage) and apoptosis inhibition		
	Vitamin E	AD	Preclinical in vitro, PMC4333972 [193] and in vivo PMC4537756 [194] and PMID: 29656360 [195]	Vit.E on astrocytes treated with glutamate: mitochondrial injuries recovering (MMP stabilization and lipid peroxidation reduction)	Epidemiological studies information is collected in the following work PMC6645610 [196]	Results insufficient. Additional studies on AD patients are needed
				Vit.E in aged mice: increase of TFAM, MMP, and ATP levels		
				Vit.E on APP/PS1 mice: cognitive and behavioral performances improvement (Aß accumulation prevention, oxidative stress reduction)		
	Vitamin E	ALS	Preclinical in vivo PMID: 8967745 [197]	Vit.E determines ALS delay onset and slows its progression	Clinical trials information is collected in the following work PMC7016185 [151]	Several clinical studies have shown conflicting outcomes in slowing ALS onset and progression, but further studies are needed

Mito-Apo = Mito-Apocynin; NAC = N-Acetylcysteine; OHDA = hydroxydopamine; SS = Szeto-Schiller; TH = Thyrosine hydroxylase; Vit. = Vitamin.

**Table 3 ijms-21-03719-t003:** Compounds acting on mitochondrial biogenesis and permeability. The table shows the promising experimental results obtained in vitro and/or in vivo after the administration of PGC-1α (to enhance mitochondria biogenesis) and Olesoxime (to regulate mitochondria permeability). The latter one already gave promising results in SMA patients. For clinical trial ID we referred to https://clinicaltrials.gov.

Therapeutic Function	Drug / Molecule	Pathology	Preclinical Studies / PMCID	Preclinical Results	Clinical Trials / Trial ID	Clinical Results
**Mitochondrial biogenesis**	PGC-1α	PD	Preclinical in vitro and in vivo PMC4293280 [202]	PGC-1α restoration in a cell culture model for α-synuclein oligomerization		
				In A30P α-syn transgenic animals: α-synuclein oligomerization reduction		
**Mitochondrial permeability**	Olesoxime	HD	Preclinical in vitro and in vivo PMID: 31283931 [206]	Mitochondrial membrane stabilization; cognitive and behavioral improvement		
	Olesoxime	PD	Preclinical in vitro and in vivo PMID: 31283931 [206]	Mitochondrial activity enhancement and apoptosis inhibition		
	Olesoxime	SMA	Preclinical in vivo PMC4033913 [204]	Lifespan elongation	Clinical trial Phase 2 Completed PMID: 31283931 [206] NCT01302600	Efficacy in motor improvement and safety have been confirmed
						It could be administered in combinatorial therapy

**Table 4 ijms-21-03719-t004:** Compounds acting on mitochondrial bioenergetics. The table lists compounds involved in mitochondrial bioenergetics enhancement, providing an overview of preclinical and clinical studies. For clinical trial ID we referred to https://clinicaltrials.gov.

Therapeutic Function	Drug / Molecule	Pathology	Preclinical Studies / PMCID	Preclinical Results	Clinical Trials / Trial ID	Clinical Results
**Mitochondrial bioenergetics**	NAD	AD	Preclinical in vitro and in vivo PMC6716473 [139]	Mitochondrial bioenergetics and dynamics enhancement and mitophagy stimulation; cognitive functions improvement	Clinical trial information is collected in the following work PMID:15134388 [207]	Lower cognitive impairment than patients treated with placebo
	Nicotinamide Riboside	AD	Preclinical in vivo PMC7016185 [151]	Learning and memory improvement (synaptic plasticity amelioration, neurogenesis enhancement, and apoptosis reduction)		
	Nicotinamide Riboside	ALS	Preclinical in vitro PMC4865928 [208]	Protection against oxidative stress		
	Nicotinamide Riboside	PD	Preclinical in vitro PMID: 29874584 [64]	Mitochondrial biogenesis and bioenergetics enhancement; MMP reduction; downregulation of ROS formation		
	Triheptanoin	ALS	Preclinical in vivo PMC5001695 [211]	Motor symptoms onset delay in SOD1^G93A^ mice thanks to mitigation of motor neuron loss		
	Triheptanoin	HD	Not found		Clinical trial Phase 2 completed PMC4336068 [210] NCT0188206	Brain metabolic profile enhancement

NAD = nicotinamide adenine dinucleotide.

**Table 5 ijms-21-03719-t005:** Compounds targeting simultaneously different mitochondrial dysfunctions. The table lists the molecules that have been tested in preclinical studies or in clinical trials and that regulate several mitochondrial processes impaired in neurodegenerative diseases. For clinical trial ID we referred to https://clinicaltrials.gov.

Therapeutic Function	Drug / Molecule	Pathology	Preclinical Studies / PMCID	Preclinical Results	Clinical Trials / Trial ID	Clinical Results
**Antioxidant, mitochondrial dynamics, permeability and bioenergetics, MMP stabilization, and apoptosis inhibition**	3-N-butylphthalide	AD	Preclinical in vitro and in vivo PMID: 30103000 [228]	In vitro: neuroprotection (apoptosis reduction and neuronal proliferation)		
				In vivo: cognitive impairment amelioration (synaptic protection, apoptosis inhibition, and antioxidant activity)		
	3-N-butylphthalide	ALS	Preclinical in vivo PMID: 30103000 [228]	Lifetime extension and motor performances improvement (motor neuron loss reduction)		
	3-N-butylphthalide	PD	Preclinical in vitro and in vivo PMID: 30103000 [228] PMID: 21524431 [229]	Neuroprotection (oxidative stress mitigation, MMP stabilization, and mPTP opening prevention)	Randomized controlled trial information is collected in the following work PMC6447885 [230] (ChiCTR1800018892)	Motor and sleep quality improvement
**Antioxidant, mitochondrial biogenesis, bioenergetics, permeability, and dynamics**	Curcumin	AD	Preclinical in vitro and in vivo PMC6716473 [139]	Oxidative stress reduction, mitochondrial biogenesis and bioenergetics enhancement, and MMP stabilization		
	Curcumin	ALS	Preclinical in vitro PMC7016185 [151]	Cytotoxicity reduction (antioxidant activity)	Clinical trials information is collected in the following work PMC7016185 [151]	Lifespan prolongation and delaying diseases progression but further studies on different delivery methods are needed
**Antioxidant, mitochondrial bioenergetics, and MMP stabilization**	Epigallocatechin-Gallate	AD	Preclinical in vitro and in vivo PMC6716473 [139]	In vitro: antioxidant activity and MMP restoration		
				In vivo: improvement of cognitive functions in rats injected with streptozotocin		
**Antioxidant and apoptosis inhibition**	Epigallocatechin- Gallate	ALS	Preclinical in vitr, and in vivo PMC7016185 [151]	In vitro: oxidative stress and lipid peroxidation reduction; apoptosis inhibition		
				In vivo: motor performances enhancement (increase of survival signal and reduction of death signal)		
**Antioxidant, mitochondrial bioenergetics, MMP stabilization, and apoptosis inhibition**	Flavonoids	ALS	Preclinical in vitro in vivo PMC7016185 [151]	In vitro: antioxidant activity		
				In vivo: motor performances improvement (prevention of MN loss)		
**Antioxidant and apoptosis inhibition**	N-Methyl,N-propynyl-2-phenylethylamine	PD	Preclinical studies in vivo PMID: 26563498 [226]	MPPE in MPTP-treated mice: neuroprotection (increase of Complex I activity and UCP-2 expression and antiapoptotic activity) and motor function enhancement		
**Antioxidant, mitochondrial dynamics and bioenergetics, MMP stabilization, and apoptosis inhibition**	Quercetin	AD	Preclinical in vitro and in vivo PMC6716473 [139]	In vitro: oxidative stress reduction and apoptosis inhibition		
				In vivo: cognitive functions improvement; antioxidant activity, MMP and mitochondrial morphology restoration, ROS reduction, ATP levels increase, and apoptosis inhibition		
**Antioxidant and mitochondrial permeability**	R(+) and S(-) Pramipexole	PD	Preclinical studies in vitro and in vivo PMID: 9648878 [224] PMID: 16407457 [225]	Neuroprotection (reduction of ROS generation and mPTP opening prevention)		
**Antioxidant, mitochondrial biogenesis and bioenergetics, MMP stabilization, and apoptosis inhibition**	Resveratrol	AD	Preclinical in vitro, and in vivo PMC6716473 [139]	In vitro: antioxidant activity, MMP restoration, apoptosis inhibition and mitophagy stimulation	Clinical trial phase Completed PMC5234138 [121] NCT0150485	Cognitive decline mitigation
				In vivo: memory loss prevention		
**Antioxidant, mitochondrial biogenesis and bioenergetics, MMP stabilization, and apoptosis inhibition**	Resveratrol with Glucose and Malate	AD	Not found		Clinical trial phase 3Completed PMC6240843 [222] NCT0067843	Safety and tolerability of low doses are confirmed
**Antioxidant, mitochondrial biogenesis and bioenergetics, MMP stabilization, and apoptosis inhibition**	Resveratrol	ALS	Preclinical in vivo PMC3996124 [220]	Resveratrol in SOD1^G93A^ ALS mice: delay in pathology onset and progression and lower and upper MNs preservation; increase of mitochondria biogenesis and regulation of autophagic flux		
**Antioxidant, mitochondrial dynamics, MMP stabilization, and apoptosis inhibition**	Tauroursodeoxycholic acid	ALS	Preclinical in vitro PMID: 24848512 [234]	Glycine-conjugated UDCA exerts an antiapoptotic activity on NSC34 cells carrying G93A mutation	Clinical trial Phase 2 Completed PMC5024041 [236] NCT0087760	Safety and disease progression decline
	Ursodeoxycholic acid	ALS	Preclinical in vitro PMID: 24848512 [234]	Already described for TUDCA	Randomized, non-controlled trial, PMC6817734 [237]	Safety and tolerability are confirmed
	Ursodeoxycholic acid	ALS	Preclinical in vitro PMID: 24848512 [234]	Already described for TUDCA	Clinical trial Phase 3, PMC6817734 [237]	ALS progression decline
	Ursodeoxycholic acid	AD	Preclinical studies in vitro PMC6193139 [235]	UDCA on fibroblasts from AD patients: MMP restoration involving Drp1		
	Tauroursodeoxycholic acid	HD	Preclinical in vivo PMC125009 [233]	Motor and sensory improvement on R6/2 mice (neuroprotection)		
	Ursodeoxycholic acid	PD	Preclinical in vitro and in vivo PMID: 30219247 [157]	Apoptosis inhibition; motor performances enhancement (low striatal dopamine decline)		
**MMP stabilization and apoptosis inhibition**	Wogonin	AD	Preclinical in vitro and in vivo PMC5478820 [219]	Wogonin on Tet-On A*β*_42_-GFP SH-SY5Y neuroblastoma cells: MMP stabilization and apoptosis inhibition		
				Wogonin on 3xTg mice: cognitive functions improvement (neuroprotective and neurotrophic activity)		

EGCG = Epigallocatechin-Gallate; MPPE = N-Methyl,N-propynyl-2-phenylethylamine; NBP = 3-N-butylphthalide; Tg = transgenic; UDCA = ursodeoxycholic acid.

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
