# Peer review of "Mitochondrial Dysfunctions: A Red Thread across Neurodegenerative Diseases"

_ijms, 2020, doi:10.3390/ijms21103719_

Round 1
Reviewer 1 Report
The manuscript "Mitochondrial dysfunctions: a red thread across neurodegenerative diseases" addresses an important issue in neurodegenerative diseases, i.e. the pathogenic role of mitochondria alterations. Authors start describing the “mitochondrial switch” in neurodegeneration, than they focus on the most common neurodegenerative diseases and finally they make a comprehensive review of potential tehrapeutic approaches to counteract neurodegeneration targeting mitochondrial dysfunctions. The manuscript is well-organized and well-written. Two very nice figures help the reader to better inderstand the topic. Tables are very helpful too. However, to complete the review, authors should also add some information on mitochondrial dysfunctions in glial cells activation and neuroinflammation, including the terapeutic approaches. Furthermore, a more complete review of mitophagy acterations should be useful.
Author Response
We thank the Reviewer for his/her comments and for the suggestions.
The points highlighted have been addressed all along the manuscript and are evidenced in yellow in the track change version.
We revised the manuscript by deepening on the concept of both mitophagy and neuroinflammation. In particular, we addressed these important aspects of neurodegeneration in the paragraph 2 entitled: The “mitochondrial switch” and its impact on neurodegeneration.
Moreover, the roles of neuroinflammation and mitophagy have been highlighted in the paragraphs corresponding to each neurodegenerative disorder.
In the section entitled “Therapies targeting mitochondria” we also indicated when the effects of the compounds reviewed were able to limit or control these processes.
Finally, we included neuroinflammation (by adding activated microglia and astrocytes) among the aspect participating in the mitochondrial switch in Figure 2.
We thank the reviewer for the observations and we refined and appropriately added some references to the literature for being more exhaustive regarding neuroinflammation and mitophagy. In the list, the added references are highlighted in yellow; we integrated 24 new references.
Please see also the attachment for cover letter and Rev1 comments.

Reviewer 2 Report
General comments to the paper entitled: Mitochondrial dysfunctions: a red thread across neurodegenerative diseases
The paper is an excellent review article. It covers all aspects of mitochondrial functions, regarding the specificity of the dysfunction in different neurodegenerative diseases. The cited 214 papers summarize all the basic knowledge well excepted: mitochondrial dysfunction is one of the first events which leads to pathological conditions: diabetes, inflammatory diseases, cancer, neurogenerative diseases. It is also well recognized that oxidative stress, disruption of mitochondrial membrane, decrease of ATP synthesis, metabolic alteration, mutation on superoxide dismutase, disbalance between the increasing radicals and antioxidant capacity are typical phenomena of different neurodegenerative diseases. The focus is on the finding the common pathogenic processes, which has not been found.
I strongly suggest the authors to add one more paragraph to review articles discussing the possible effect of deuterium on mitochondrial functions, in living organisms. Recently several papers were published to show the effect of deuterium depletion on cancer cells, tumor growth, diabetes, aging, depression, long-term memory, ATP synthesis, oxidative stress, SOD induction, antioxidant effect. The data clearly indicate the importance of deuterium/hydrogen ratio and the key role of the well-functioning mitochondria which is able to produce deuterium-depleted metabolic water in healthy cells.
I think the paper will be completed by adding this new paragraph, which can be the “red thread”.
Below, you can find some relevant papers.
Kotyk A. Dvorakova M. Koryta J. (1990) Deuterons cannot replace protons in active transport processes in yeast. FEBS Letters 264(2). 203-205
Richard J. Robins, Isabelle Billault, Jia-Rong Duan, S´ebastien Guiet, S´ebastien Pionnier & Ben-Li Zhang, (2003) Measurement of 2H distribution in natural products by quantitative 2H NMR: An approach to understanding metabolism and enzyme mechanism? Phytochemistry Reviews 2: 87–102, 2003).
Kirk Goodall (2003) The Role of Deuterium in DNA Degradation. Anti-Aging Medical News3:7-31.
A. Pomytkin and O. E. Kolesova (2006) Relationship between Natural Concentration of Heavy Water Isotopologs and Rate of H2O2 Generation by Mitochondria. Bulletin of Experimental’noi Biologii i Meditsiny, 142(11):514-516.
Abdullah Olgun (2007) Biological effects of deuteration: ATP synthase as an example. Theoretical Biology and Medical Modelling, 4:9
Cristian Mladin, Alin Ciobica, Radu Lefter, Alexandru Popescu, Walther Bild (2014) Deuterium-depleted water has stimulating effects on long-term memory in rats. Neuroscience Letters 583, 154-158.
Tatyana Strekalova, Matthew Evans, Anton Chernopiatko, Yvonne Couch, Joao Costa-Nunes, Raymond Cespuglio, Lesley Chesson, Julie Vignisse, Harry W. Steinbusch, Daniel C. Anthony, Igor Pomytkin, Klaus-Peter Lesch (2015) Deuterium content of water increases depression susceptibility: The potential role of a serotonin-related mechanism. Behavioural Brain Research 277, 237-244.
Oleg I. Kit, Alla I. Shikhliarova, Galina V. Zhukova, Stepan S. Dzhimak, Mikhail G. Barisev, Tatiana A. Kurkina, Elena A. Shirnina, Tatiana P. Prostasova. (2017) Relation of antistress and geroprotectuive effects of deuterium depleted water in aging female rats. Original Research DOI:10.12710/cardiometry.2017.35-42.
Youping Zhou, Benli Zhang, Hilary Stuart-Williams, Kliti Grice, Charles H. Hocart, Arthur Gessler, Zachary E. Kayler, Graham D. Farquhar (2018) On the contributions of photorespiration and compartmentation to the contrastingintramolecular 2H profiles of C3 and C4 plant sugars Phytochemistry 145: 197-206. doi: 10.1016/j.phytochem.
Xuepei Zhang, Massimiliano Gaetani, Alexey Chernobrovkin, Roman A.Zubarev (2019) Anticancer effect of deuterium depleted water-redox disbalance leads to oxidative stress. Mol Cell Proteomics 18(12):2373-2387. doi: 10.1074/mcp.RA119.001455.
Tetiana Halenova, Igor Zlatskiy, Anton Syroeshkin, Tatiana Maximova and Tatiana Pleteneva. (2020) Deuterium-Depleted Water as Adjuvant Therapeutic Agent for Treatment of Diet-Induced Obesity in Rats. Molecules 23,25
Alexander Basov, Liliia Fedulova, Mikhail Baryshev and Stepan Dzhimak. (2019) Deuterium-Depleted Water Infulence on the Isotope 2H/1H Regulation in Body and Individual Adaptation. Nutrients 11,1903 1-19.
Gábor Somlyai, Gábor Jancsó, György Jákli, Kornélia Vass, Balázs Barna, Viktor Lakics and Tamás Gaál (1993) Naturally occurring deuterium is essential for the normal growth rate of cells. FEBS Lett. 317, 1-4.
Laskay G. Somlyai G. Jancsó G.: Reduced deuterium concentration of water stimulates O2-uptake and electrogenic H+-efflux in the aquatic macrophyte Elodea Canadensis. Jpn. J. Deuterium Sci. 2001. 10. 17-23
Author Response
We thank the Reviewer for suggesting this mechanism related to oxidative stress and playing an important role in many disorders, not only neurodegeneration.
We added the process of deuteronation among the mechanisms altered in neurodegenerative diseases. Revisions can be found in the paragraph entitled: The “mitochondrial switch” and its impact on neurodegeneration.
We are also grateful to the Reviewer for the articles he/she suggested to mention: accordingly, we refined and appropriately enriched the references’ list; we integrated 24 new references.
PLEASE see the attachment, for cover letter and comments Rev 2.
